# FDG PET/CT and Histopathologic Characteristics of Low-Grade Appendiceal Mucinous Neoplasm with Pseudomyxoma Peritonei

**DOI:** 10.3390/diagnostics15243198

**Published:** 2025-12-15

**Authors:** Li-Wei Liu, Hung-Pin Chan, Daniel Hueng-Yuan Shen, Chang-Chung Lin

**Affiliations:** Department of Nuclear Medicine, Kaohsiung Veterans General Hospital, Kaohsiung City 813, Taiwan

**Keywords:** ^18^F-FDG PET/CT, pseudomyxoma peritonei, low-grade appendiceal mucinous neoplasm

## Abstract

A 50-year-old woman presented a rapidly proliferative cystic lesion in the left pelvis as displayed by sonography. Exploratory laparoscopy with omental biopsy suggested pseudomyxoma peritonei (PMP). An ^18^F-FDG PET/CT revealed scalloping of the liver surface with an associated extrahepatic lesion exhibiting moderate FDG avidity and strand-like FDG uptake within a left-sided septated ovarian cyst. The diagnosis of low-grade appendiceal mucinous neoplasms was proven after cytoreductive surgery. This case provided important imaging features on ^18^F-FDG PET/CT that directed the pre-operative diagnosis and the initiation of treatment.

**Figure 1 diagnostics-15-03198-f001:**
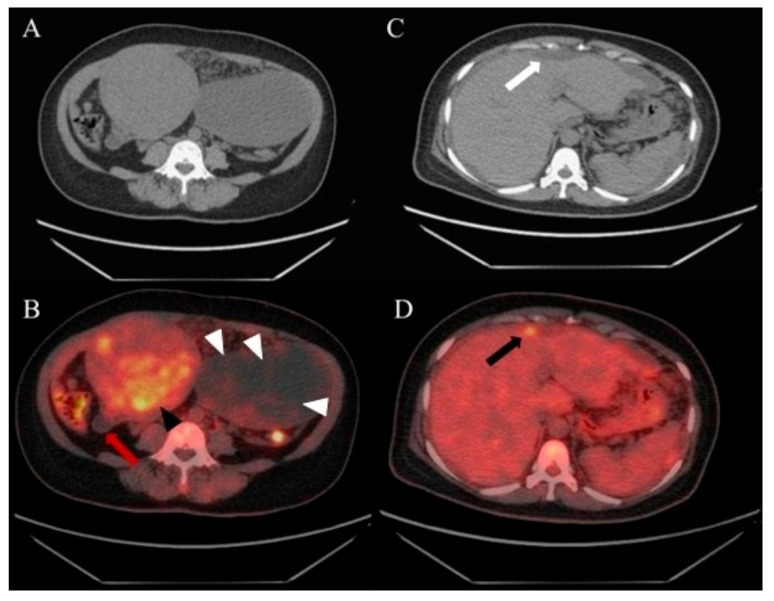
A 50-year-old woman complaining of pelvic heaviness for months and abdominal ultrasounds revealed a mixed-echogenic uterine mass and a cystic lesion with hyperechoic septa. Serum CA-199, CEA, and CA-125 level were elevated. Exploratory laparoscopy and biopsy confirmed a fundal myoma, left ovarian cyst, and mucinous material disseminated throughout the peritoneal and pelvic cavities, with deposits found involving ovaries, spleen, liver, and the peritoneal surface of the diaphragm. Omental biopsy indicated pseudomyxoma peritonei (PMP), suspected origins from the appendix or ovaries. To evaluate the systemic involvement, the ^18^F-FDG PET/CT was performed and displayed, no apparent supra-diaphragmatic FDG uptake, a uterine mass with moderate FDG uptake ((**A**), unenhanced CT image; (**B**), indicated by a black arrowhead), consistent with a diagnosis of uterine myoma, poorly FDG-avid, strand-like appearance within the left ovarian cystic lesion ((**B**), white arrowheads), physiologic FDG uptake in the compressed descending colon posterior to the cystic ovarian lesion (**B**), and a cystic appendix showing no FDG uptake ((**B**), red arrow). Note the CT showed scalloping of the liver surface ((**C**), white arrow) accompanied by an uneven FDG-avid extrahepatic lesion ((**D**), black arrow). Several FDG-avid areas were also observed on the omentum.

**Figure 2 diagnostics-15-03198-f002:**
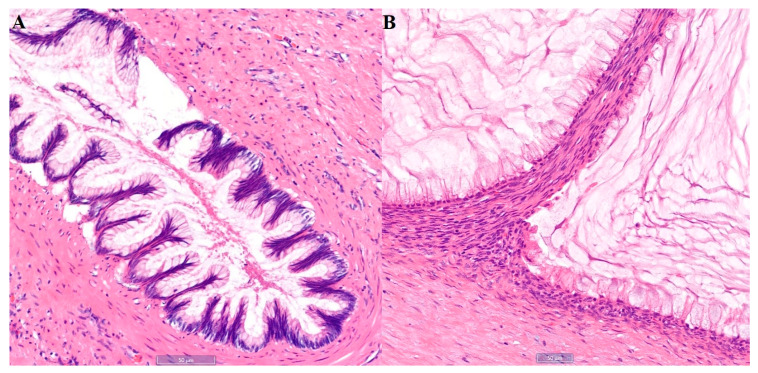
The patient elected to receive cytoreductive surgery with Hyperthermic Intraperitoneal Chemotherapy (HIPEC), which was performed shortly after PET/CT imaging. The pathology confirmed the diagnosis of uterine leiomyoma and also low-grade appendiceal mucinous neoplasm (LAMN) involving the appendix (**A**), left ovary (**B**) and omentum. Histological examination revealed low-grade columnar cells with apical intracellular mucin, and extracellular mucin pooling. Immunohistochemistry showed SATB2, CDX2, CK20 positivity in the neoplastic cells, and PAX8 and CK7 negativity, pointing to an appendiceal origin. LAMN is a rare appendiceal neoplasm, usually affecting adults in their sixth decade of life. Along with other appendiceal mucinous neoplasms (AMN), LAMN make up around 1% of appendectomy specimens [1]. In rare cases, peritoneal dissemination of LAMN via appendiceal rupture eventually results in PMP. With incidence of around 1–2/million people per year [2,3], PMP is a clinical syndrome caused by mucin buildup within the peritoneal cavity produced by neoplastic cells deposited on the peritoneum and organ surfaces [4]. The etiology is overwhelmingly appendiceal. Its clinical course is indolent, with most patients only experiencing non-specific abdominal symptoms, such as abdominal distention and new-onset hernia, after the formation of significant mucinous ascites [5,6]. CT is currently one of the diagnostic methodologies for LAMN and PMP [1,5]. AMN confined to the appendix are commonly associated with soft tissue irregularities and calcifications of the appendiceal wall. PMP can present with “scalloping” of the liver and spleen, which indicates the mucinous nature of the ascites [7]. However, ^18^F-FDG PET/CT findings in AMN and PMP are rarely published, probably due to relatively low sensitivity for low-tumor-volume diseases and mucinous lesions [5]. However, it might be correlated between FDG-avidity and histological grade of PMP and AMN on retrospective studies [4,8]. This case demonstrates a unique strand-like FDG appearance within the left ovarian cystic lesion on ^18^F-FDG PET/CT. Although the cystic appendix shows no FDG uptake, an uneven FDG-avid extrahepatic lesion was incidentally detected due to the irregular liver surface. Additionally, the uterine FDG uptake is attributed to the activity of the patient’s myoma. Given the inconsistent FDG uptake and imaging features across different organs, such cases are rarely reported in the literature on ^18^F-FDG PET/CT.

## Data Availability

The original contributions presented in this study are included in the article. Further inquiries can be directed to the corresponding author.

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
