# Peer review of "FDG PET/CT and Histopathologic Characteristics of Low-Grade Appendiceal Mucinous Neoplasm with Pseudomyxoma Peritonei"

_diagnostics, 2025, doi:10.3390/diagnostics15243198_

Round 1
Reviewer 1 Report
Comments and Suggestions for Authors
Liu and colleagues present a case report on the use of PET/CT in pseudomyxoma.
Was there any histological/clinical correlation between the avid PET areas (ovary & liver surface lesion)? i.e. did they patient have a Krukenberg tumour?
Presenting the fused PET/CT imaging is often helpful, if available
We find that PET/CT in low grade PMP is of limited value. It can however help in identifying high grade disease. Have the authors any experience with this?
Author Response
Dear Reviewer,
Comment 1: Was there any histological/clinical correlation between the avid PET areas (ovary & liver surface lesion)? i.e. did they patient have a Krukenberg tumour?
Response 1: We thank the reviewer for the comment. The neoplastic cells involving the ovaries, appendix, and omentum were histologically and immunohistochemically identical, showing positive SATB2, CDX2, and CD20, and negative for CD7 and PAX8. Unfortunately, the liver lesion was not submitted for pathological diagnosis. We have made changes to our manuscript to better demonstrate the histological/clinical correlation [Lines 40-42: Histological examination revealed low-grade columnar cells with apical intracellular mucin, and extracellular mucin pooling. Immunohistochemistry showed SATB2, CDX2, CK20 positivity in the neoplastic cells, and PAX8 and CK7 negativity, pointing to an appendiceal origin. ]
Comment 2: Presenting the fused PET/CT imaging is often helpful, if available
Response 2: We thank the reviewer for this comment. We have updated the manuscript with a color version of the image in Figure 1 for better visualization of the FDG avidity of the described lesions.
Comment 3: We find that PET/CT in low grade PMP is of limited value. It can however help in identifying high grade disease. Have the authors any experience with this?
Response 3: We thank the reviewer for this comment. We have encounter similar cases in the past on rare occasions. In our opinion, PET/CT was able to provide diagnostic value for locating areas with higher tumor burden in low-grade disease.
Reviewer 2 Report
Comments and Suggestions for Authors
The Authors have presented a case of low-grade appendiceal mucinous neoplasm involving the ovaries and omentum (pseudomyxoma peritonei), undergoing FDG PET-CT for incidental finding of a pelvic mass of uterine/ovarian origin.
The case is overall well described, although some minor issue are to be corrected, especially in CT and PET-CT images:
1) it should be desirable to provide color-images since some findings are not easy to detect on b&w ones: I had some troubles in finding the red arrow pointing to the cystic appendix in subfigure B, and low-uptake FDG findings are easier to be noticed on fused color-images or on PET-alone images (e.g. in subfigure D it seems there is faint FDG uptake along the lateral surface of spleen, although less intense than that displayed along the surface of liver)
2) some focal uptake sites should be described better, particularly the focal intense FDG collection in subfigure B just behind the left cystic mass: is it physiological due to diffuse uptake in the descending colon? is it anything else?
3) was the focal uptake along the anterior surface of liver the only intense site of FDG uptake along the peritoneum? and in this case, was it further investigated (e.g. using contrast-enhanced CT or MRI)?
4) MIP image should be added to confirm the absence of supra-diaphragmatic involvement
Best regards
Author Response
Comment 1: It should be desirable to provide color-images since some findings are not easy to detect on b&w ones: I had some troubles in finding the red arrow pointing to the cystic appendix in subfigure B, and low-uptake FDG findings are easier to be noticed on fused color-images or on PET-alone images (e.g. in subfigure D it seems there is faint FDG uptake along the lateral surface of spleen, although less intense than that displayed along the surface of liver)
Response 1: We thank the reviewer for this comment. We have updated the manuscript with the color version of Figure 1 for better visualization.
Comment 2: Some focal uptake sites should be described better, particularly the focal intense FDG collection in subfigure B just behind the left cystic mass: is it physiological due to diffuse uptake in the descending colon? is it anything else?
Response 2: We agree and thank the reviewer for this comment. The focal intense FDG uptake in subfigure B was likely due to physiological uptake of a compressed descending colon, as no gross involvement by the neoplasm was observed during her cytoreductive surgery. We have made changes to our manuscript to reflect the negative findings [Lines 33-34: physiologic FDG uptake in the compressed descending colon posterior to the cystic ovarian lesion (1-B),].
Comment 3: Was the focal uptake along the anterior surface of liver the only intense site of FDG uptake along the peritoneum? and in this case, was it further investigated (e.g. using contrast-enhanced CT or MRI)?
Response 3: We thank the reviewer for this comment. There were multiple sites with intense FDG uptake on the omentum surface correlating with the gross involvement seen during the patient's surgical intervention. Unfortunately, the patient elected to received cytoreductive surgery shortly after receiving the results of the PET/CT, which prevented any additional imaging studies to be done before the involved sites in the peritoneal cavity were disrupted. We have made changes to our manuscript to reflect the clinical decision [Lines 36-37: Several FDG-avid areas were also observed on the omentum (not shown). ;Lines 38-39: The patient elected to receive ... shortly after PET/CT imaging].
Comment 4: MIP image should be added to confirm the absence of supra-diaphragmatic involvement
Response 4: We thank the reviewer for this comment. We did not find areas of abnormal FDG uptake in the supra-diaphragmatic region in both our MIP and PET/CT images, thus we elected to not include them in our manuscript. We have made changes to our manuscript to reflect the decision [Lines 31:, no apparent supra-diaphragmatic FDG uptake, ].